# An Improved Lightweight Network Using Attentive Feature Aggregation for Object Detection in Autonomous Driving

Priyank Kalgaonkar and Mohamed El-Sharkawy *

Department of Electrical and Computer Engineering, Purdue School of Engineering and Technology, Indianapolis, IN 46254, USA; pkalgaon@purdue.edu
* Correspondence: melshark@purdue.edu

**Abstract:** Object detection, a more advanced application of computer vision than image classification, utilizes deep neural networks to predict objects in an input image and determine their locations through bounding boxes. The field of artificial intelligence has increasingly focused on the demands of autonomous driving, which require both high accuracy and fast inference speeds. This research paper aims to address this demand by introducing an efficient lightweight network for object detection specifically designed for self-driving vehicles. The proposed network, named MobDet3, incorporates a modified MobileNetV3 as its backbone, leveraging its lightweight convolutional neural network algorithm to extract and aggregate image features. Furthermore, the network integrates altered techniques in computer vision and adjusts to the most recent iteration of the PyTorch framework. The MobDet3 network enhances not only object positioning ability but also the reusability of feature maps across different scales. Extensive evaluations were conducted to assess the effectiveness of the proposed network, utilizing an autonomous driving dataset, as well as large-scale everyday human and object datasets. These evaluations were performed on NXP BlueBox 2.0, an advanced edge development platform designed for autonomous vehicles. The results demonstrate that the proposed lightweight object detection network achieves a mean precision of up to 58.30% on the BDD100K dataset and a high inference speed of up to 88.92 frames per second on NXP BlueBox 2.0, making it well-suited for real-time object detection in autonomous driving applications.

**Keywords:** MobileNetV3; lightweight; object detection; autonomous driving; PyTorch; NXP

## 1. Introduction

Deep learning techniques and strategies are a subset of machine learning algorithms in the field of Artificial Intelligence (AI), that use many layers of nonlinear processing units (neurons) for feature extraction, representations, and transformations. Due to the abundance of extensive real-world data and enhanced computational capabilities over the last ten years, Convolutional Neural Network (CNN), which is a specific type of Deep Neural Network (DNN), has gained significant traction among researchers and engineers. It has emerged as a favored option for designing and implementing innovative techniques and algorithms in computer vision systems. These systems leverage multiple layers of neurons to carry out intricate tasks, such as image classification and object detection. These algorithms may then be trained for pattern analysis in an unsupervised manner, or for image classification or object detection in a supervised manner [1].

Although interest in deep learning and AI has gained momentum in recent years, its roots can be traced back to as far as the 1950s, inspired by neuroscientific research of the operation of neurons in the human brain, when a simple supervised linear model of Artificial Neural Networks (ANN) associated with input $x$ and output $y$ was proposed [2], and the first supervised, deep, feed-forward, multi-layer ANN was proposed by Alexey G. Ivakhnenko and V. G. Lapa [3].

In 2000, Aizenberg et al. introduced the term 'Deep Learning' to ANNs [4]. It was in this decade where the topic of deep learning caught the eye of researchers and engineers and sparked an interest around the world following Hinton et al. publishing an article [5] endorsed by the Canadian Institute for Advanced Research (CIFAR). Similar research in deep learning [6,7] quickly followed, which also demonstrated that deep networks outperform shallow networks and consequently popularized the use of the term 'Deep Learning' as it is known today.

Following the groundbreaking achievements in deep learning in the 2000s, in 2010, the United States government's Defense Advanced Research Projects Agency (DARPA), Microsoft, and Google sponsored research to develop novel technologies [8,9]. In 2012, AlexNet, a deep CNN proposed by Alex Krizhevsky, Ilya Sutskever, and Geoffrey Hinton, won the ImageNet LSVRC (Large Scale Visual Recognition Challenge) competition, leading to many AI-related revolutions and applications utilizing CNNs and GPUs leveraging large-scale datasets. In recent times, there has been a notable surge in the utilization of computer vision, leading to a demand for the creation of new artificial neural network (ANN) models and methodologies. These are aimed at accomplishing intricate tasks, such as image classification and object detection, using Convolutional Neural Networks (CNNs) with multiple layers and neurons. The objective is to mimic the innate visual perception exhibited by both humans and animals.

In this paper, a novel modern object detector called MobDet3 is proposed, which is designed to perform object detection with attentive feature aggregation for autonomous driving using MobileNetV3, a lightweight CNN designed specifically to run on edge devices with constrained computational resources. The proposed work can be summarized as follows:

1. A new lightweight object detector capable of being trained on a single GPU system.
2. Incorporates modified versions of cutting-edge algorithms and methodologies, such as FPN [10], PAN [11], YOLO [12], and GIoU_Loss [13].
3. Thorough training and ablation studies have been conducted on two significant multi-class datasets, namely BDD100K [14] and Microsoft COCO [15].
4. Trained weights have then been deployed on NXP BlueBox 2.0 [16] to perform real-time vehicular object detection and Frame-Per-Second (FPS) evaluation.

The rest of this paper is organized as follows:

- Section 2 provides an overview of object detection models and related work.
- Section 3 describes the methodology of the proposed MobDet3 object detection network.
- Section 4 provides an overview of NXP BlueBox 2.0, an advanced automotive development platform used in the experiments.
- Section 5 evaluates the MobDet3 object detector by benchmarking it on different datasets and on NXP BlueBox 2.0.
- Section 6 concludes the paper.

## 2. Background and Literature Review

The task of an object detector is to determine categories of object instances and spatially locate them in an input image and draw bounding boxes, along with assigning class labels for all detected objects of interest, as seen in Figure 1 below.

Object detection is a computer vision technique that enables machines to identify and locate objects within an image or video. It is a crucial task in various applications, including autonomous driving, surveillance, and robotics. Object detection can be performed using both traditional computer vision approaches and deep learning techniques. Traditional approaches rely on handcrafted features and classifiers, while deep learning techniques use neural networks to learn features and classifiers directly from the data.

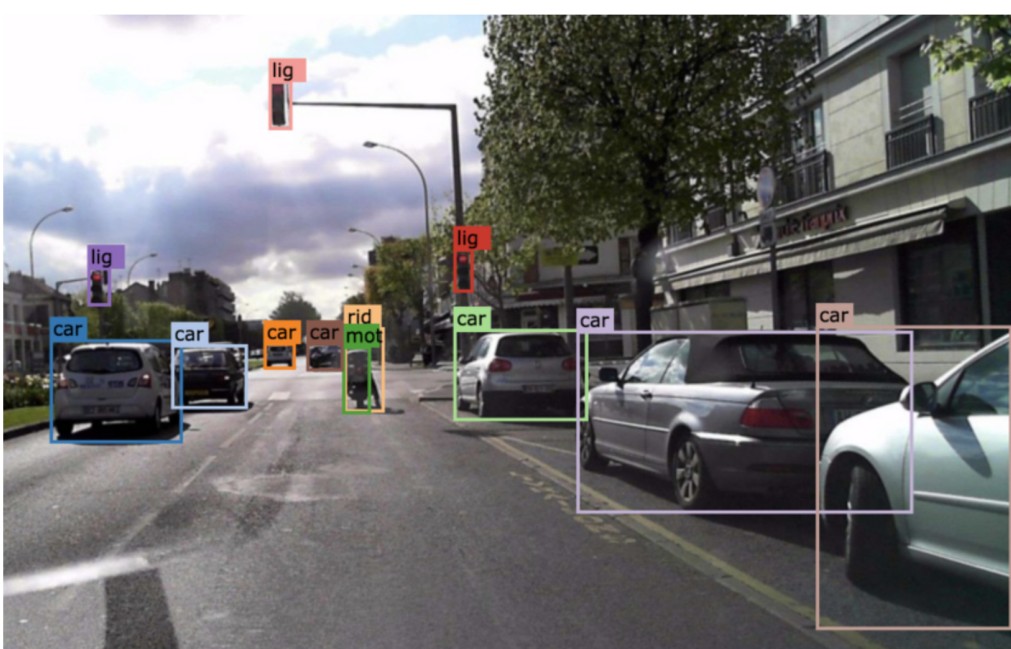

**Figure 1.** Object detection on an image from the BDD100K dataset [14].

In recent years, deep learning-based object detection methods have achieved remarkable performance improvements. One of the most popular deep learning architectures for object detection is the convolutional neural network (CNN). Early CNN-based object detection methods, such as R-CNN [17] and its variants, divided the object detection task into two stages: region proposal and classification. These methods achieved state-of-the-art performance but were computationally expensive due to the need for expensive region proposal methods.

Later, single-stage object detection methods, such as YOLO (You Only Look Once) [12] and SSD (Single Shot Detector) [18], were introduced, eliminating the need for region proposal methods and achieving faster detection speed. These methods are now widely used for real-time object detection applications, such as autonomous driving.

### 2.1. Environmental Awareness for Autonomous Vehicles: The Role of Visual Perception

Visual perception is a critical aspect of autonomous vehicles, enabling them to perceive the environment and make decisions based on that perception. Visual perception of autonomous vehicles involve the use of cameras, lidars, and radars to capture images, point clouds, and range data, respectively. The data are then processed using computer vision and machine learning algorithms to detect and classify objects, predict their motion, and estimate their pose.

Several techniques have been proposed for visual perception in autonomous vehicles. One of the most popular approaches is the use of convolutional neural networks (CNNs) for object detection and classification. CNNs have shown remarkable performance in detecting and recognizing objects and have been used in many autonomous vehicles for tasks such as traffic and pedestrian detection, traffic sign recognition, and lane detection.

Two-dimensional object detection has been a widely researched topic in the field of computer vision, and various approaches have been proposed to achieve accurate and efficient object detection. Deep learning-based approaches using Faster R-CNN [19] and YOLO [12] object detection frameworks have shown significant improvements in object detection accuracy and speed compared to traditional computer vision methods.

Challenges in 2D object detection for autonomous vehicles include dealing with occlusions, variations in lighting conditions, and diverse object appearances, in addition to constrained computational resources to process the input data received from multiple sensors, such as cameras, in real time. To overcome these challenges, there has been an on-going

demand to develop novel state-of-the-art techniques and methodologies to continuously improve existing as well as develop new and more efficient object detection algorithms.

Another important factor to consider when designing object-detecting deep neural networks (DNNs) for visual perception systems in autonomous vehicles is the framerate of the detector. The framerate, which represents the number of Frames Per Second (FPS) that the DNN can process, has a direct impact on the vehicle's stopping distance and other maneuvering actions. As a result, real-time frame rates are an important factor in evaluating such a DNN to ensure feasible vehicle control.

### 2.2. Advancing Autonomous Driving: The Significance of Real-Time Object Detection

There are two main architectures employed in object detection using deep neural networks: region-based networks (two-stage) and regression-based networks (one stage). In the two-stage framework, the initial stage involves generating region proposals that are not specific to any particular categories. These regions then undergo feature extraction using convolutional neural networks (CNN). In the second stage, category-specific classifiers are employed to determine the category labels of the proposals. However, two-stage networks generate numerous region proposals during testing, resulting in significant computational expenses. The swiftest two-stage object detector networks, such as Faster R-CNN [19] and R-FCN [20], can process images at around 50–60 frames per second (FPS).

In contrast, one-stage networks employ a single feedforward CNN network to directly predict class probabilities and bounding box offsets from complete input images. This approach eliminates the need for region proposal generation and subsequent feature resampling stages, enabling the network to be optimized end-to-end for detection performance. Although one-stage networks generally exhibit slightly lower accuracy compared to two-stage networks, they often boast real-time object detection capabilities. It is worth mentioning that the term "real-time" does not have a formally defined time limit or frame rate. Instead, a real-time computing system ensures a response within a specified deadline.

For a real-time visual perception system utilized in autonomous vehicles, the time constraint can be determined by either the camera frame rate or the distance covered by detected objects between successive frames. Due to their low frame rates, two-stage networks are not suitable for the majority of self-driving vehicle scenarios. To achieve real-time object detection in autonomous vehicles, a system should operate at frame rates exceeding 30 FPS. The prominent two-stage frameworks consist of Faster R-CNN, RFCN, FPN, and Mask R-CNN, whereas the one-stage methods primarily encompass various versions of YOLO and SSD.

### 2.3. Exploring the Inner Workings of Object Detectors

A modern object detector is constructed using three primary components: a backbone that extracts features from the input image using a strong image classifier, a neck that connects to the backbone and aggregates feature maps from various stages of the backbone to combine them, and a head that detects bounding boxes and predicts object classes. A simple visual depiction of the head, neck, and backbone of a modern object detector can be seen in Figure 2.

Typically, each module of a contemporary object detector model comprises a blend of the techniques listed below:

- Backbone: powerful image classifier (CNN) such as MobileNetv3 [21], CondenseNeXt [22], VGG16 [23], ResNet-50 [24], SpineNet [25], etc.
- Neck (feature aggregator):
  - Additional Blocks: SPP [26], RFB [27], ASPP [28], etc.
  - Path Aggregation Blocks: FPN [10], PAN [29,30] ASFF [30], etc.
- Head (to draw bounding boxes and make class predictions):
  - One-Stage (Dense) Predictions: SSD [18], YOLO [12], CenterNet [31], RetinaNet [32], etc.

○ Two-Stage (Sparse) Predictions: R-FCN [20], R-CNN [17], Fast R-CNN [33], Faster R-CNN [19], R-FCN [20], Cascade R-CNN [34], etc.

**Figure 2.** General anatomy of an object detector. The backbone module of a modern object detector extracts features from the input image at multiple resolutions using a powerful image classifier. The neck module, which is connected to the backbone, merges features from different resolutions to create a more comprehensive representation of the input image. The head module then uses the merged features to identify bounding boxes and predict object classes.

## 3. MobDet3

The following section introduces MobDet3, a novel object detection network based on the YOLOv5 [35] framework. By utilizing Attentive Feature Aggregation, MobDet3 provides an improved lightweight solution for object detection in autonomous driving applications. The network is designed to be efficient and effective, even on resource-limited embedded systems such as the NXP BlueBox 2.0 [16]. Experimental evaluations conducted on various datasets on NXP BlueBox in Section 5 confirm the network's superior performance and subsequently validate the results presented in this paper.

### 3.1. Backbone

The backbone of a modern object classifier is responsible for extracting essential features from an input image at different levels of coarseness. This allows the object classifier to detect objects of different sizes and scales. However, for resource-constrained devices, computational efficiency must also be considered, in addition to prediction accuracy. Although the YOLO family employs the Darknet CNN introduced in [36], as its backbone, leading to enhanced detection performance, this approach comes with the drawback of increased parameterization. MobDet3, on the other hand, employs a modified version of the MobileNetV3 CNN architecture as its backbone, which belongs to the family of efficient CNNs designed for mobile and embedded vision applications developed by Google researchers.

This methodology captures spatial information from each layer and efficiently transfers it to subsequent layers in a feed-forward manner. As a result, features can be extracted effectively at various resolutions. To address the vanishing-gradient problem, the proposed network utilizes depthwise separable convolution and pooling layers to fuse information from various resolutions. Figure 3 illustrates the architecture of MobileNetV3. The experiments on various datasets in Section 4 demonstrate that MobDet3 is faster and lighter and can perform object detection efficiently on resource-constrained embedded systems.

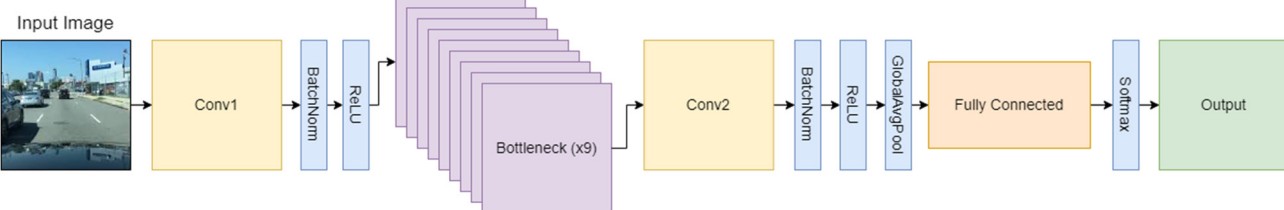

**Figure 3.** Visual representation of MobileNetV3 architecture.

The MobileNetV3 backbone employed in the proposed object detection network, MobDet3, has been optimized for lightweight performance beyond its original image classification design. This is achieved by removing the final fully connected and softmax layers, making the backbone compatible with, and facilitating its integration with the neck module, as explained in the following section.

MobileNetV3 CNN

MobileNetV3 is a lightweight CNN model that is particularly well suited for use on devices with constrained computational resources. It was developed through a platform-aware network architecture search and the implementation of the NetAdapt algorithm [37]. The MobileNetV1 [38] (MobileNet version 1) model introduced depthwise convolution, which reduced the number of parameters. MobileNetV2 [39] introduced an implementation of an expansion layer that was added to each block, creating a system of expansion-filtering-compression, visually represented in Figure 4, using three layers. This system, called the Inverted Residual Block, helped to further improve performance.

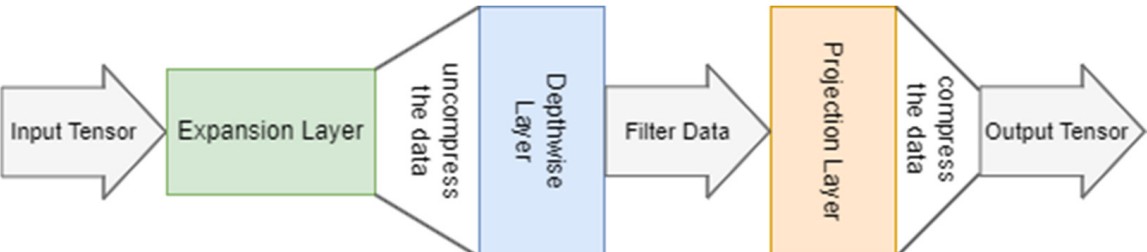

**Figure 4.** Expansion-Filtering-Compression System of MobileNetV3.

MobileNetV3 builds on the strengths of its predecessors by combining inverted residual bottlenecks from MobileNetV2 with Squeeze and Excitation (SE) blocks from Squeeze and Excitation Networks [40]. The SE blocks are included to enhance the classification-relevant feature maps and suppress those that are not useful. The residual bottlenecks in MobileNet V3 also consist of a $1 \times 1$ expansion convolution layer, followed by a $3 \times 3$ Depthwise Convolution (DWC) layer, and then a $1 \times 1$ projection layer.

MobileNetV3 incorporates depthwise separable convolutions as a more efficient alternative to conventional convolution layers. This technique effectively divides traditional convolution into separate layers, separating spatial filtering from feature generation. Depthwise separable convolutions consist of two distinct layers:

1.  Depthwise Convolution: A lightweight convolution for spatial filtering mathematically represented by Equation (1) and graphically represented in Figure 5.

$$\hat{Y}_{k,l,m} = \sum_{i,j} \hat{K}_{i,j,m} \cdot X_{k+1-1,l+j-1,m} \tag{1}$$

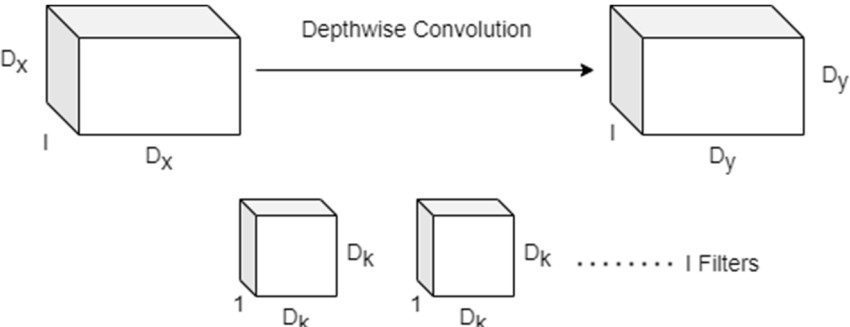

**Figure 5.** Depthwise convolution used for spatial filtering.

Here, $X$ represents the input feature map of size $D_x \times D_x \times I$, where $I$ corresponds to the number of input channels.

2. Pointwise Convolution: A more substantial 1×1 Pointwise Convolution for feature generation mathematically represented by Equation (2) and graphically represented in Figure 6.

$$Y_{k,l,n} = \sum_m \widetilde{K}_{m,n} \cdot \hat{Y}_{k-1,l-1,m} \tag{2}$$

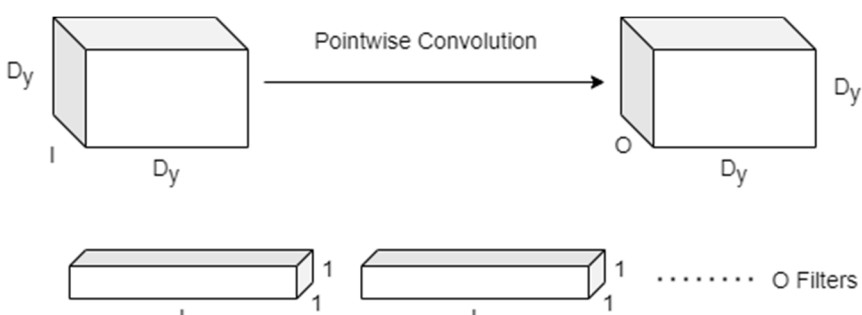

**Figure 6.** Pointwise convolution for feature generation.

Here, $Y$ represents the output feature map of size $D_y \times D_y \times O$, where $O$ corresponds to the number of output channels.

Thus, depthwise separable convolution reduces the number of parameters and computations compared to traditional convolution while maintaining high accuracy.

*3.2. Neck*

3.2.1. FPN and PAN

In contemporary object detectors, the neck module is linked to the backbone module and serves as a feature aggregator. It accomplishes this by gathering feature maps from different stages of the backbone and merging them using pyramid networks, such as Feature Pyramid Networks and Path Aggregation Network.

The proposed object detector incorporates the neck module by linking the Path Aggregation Network (PAN) to the Feature Pyramid Network (FPN). The FPN generates feature maps of varying sizes, facilitating the fusion of diverse features. However, due to the different sizes of feature maps within the pyramid, it becomes challenging to merge the bottom features with the top features. To tackle this challenge, the PAN is integrated with the FPN and upsampled by a factor of two using the nearest-neighbor approach. This enables the connection of bottom features with top features. This modified approach enhances detection performance by providing more robust positioning features from the bottom-up and stronger semantic features from the top-down. Figure 7 depicts a simple graphical representation of this technique.

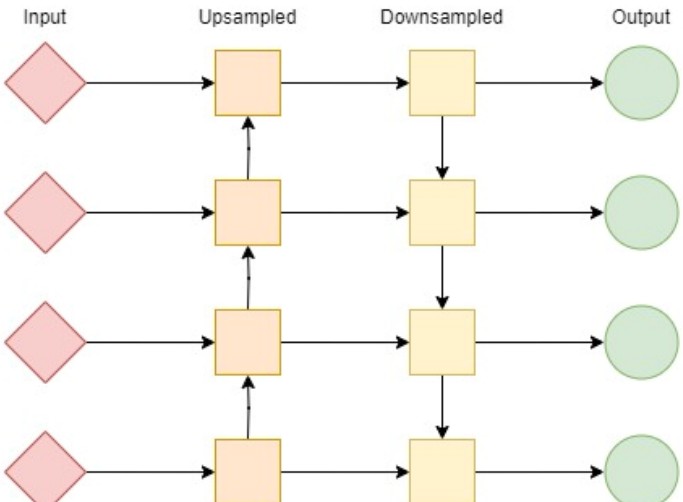

**Figure 7.** A graphical representation of PAN implementation of the neck of the proposed MobDet3. Here, input is from the backbone of the network, and output is to the head of the network.

### 3.2.2. SPP

Spatial pyramid pooling (SPP) is a pooling technique that improves the accuracy of convolutional neural networks (CNNs) by pooling the responses of every filter in each local spatial bin. This preserves the spatial information of the input image, which can be helpful for object detection and other tasks that require spatial information.

SPP is inspired by the bag-of-words approach in computer vision. In bag-of-words, images are represented as a bag of features where each feature is a descriptor of a local region of the image. SPP works similarly, but instead of using a fixed set of features, it uses features extracted by a CNN.

The SPP uses three different sizes of max-pooling operations to identify similar feature maps. This allows the SPP to be more robust to variations in input feature patterns. For example, if an object is rotated or scaled, the SPP will still be able to identify the object because it will be able to find similar feature maps in the different rotated or scaled versions of the object.

Following the max-pooling operations, the resulting outputs are flattened, concatenated, and directed to a fully connected (FC) layer that produces an output of a fixed size. This process is illustrated in Figure 8. The limitation of having a fixed-size output in CNNs is primarily attributed to the fully connected (FC) layer rather than the convolution layer. In the proposed object detection network, Spatial Pyramid Pooling (SPP) is integrated into the latter stages, where features are extracted.

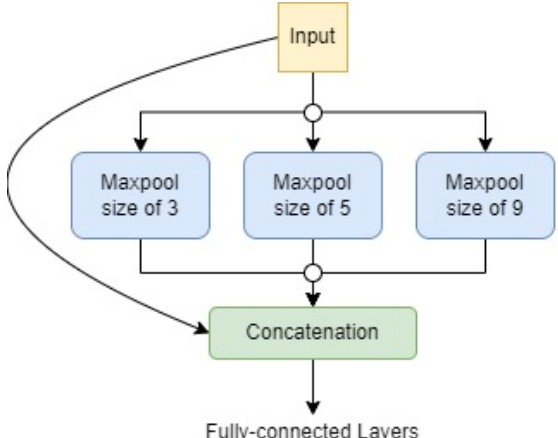

**Figure 8.** A graphical representation of the SPP implementation of the neck of the proposed MobDet3.

### 3.3. Head

In a modern object detector, the primary module is responsible for identifying bounding boxes and providing detection outcomes, including the object's class, location, size, and score. In order to accomplish accurate object detection in an input image and generate score predictions, it is common practice to employ multiple head modules in the design of object detectors. These head modules utilize a shared feature set from earlier stages in the network.

The MobDet3 object detection network proposed in this study utilizes three heads, each with a Spatial Attention Module (SAM) block for attentive feature aggregation, based on the CBAM [41]. This approach generates a spatial attention map that prioritizes important areas. Inter-spatial relationships of features are obtained using average-pooling and max-pooling operations along the channel axis, and this can be expressed mathematically as follows, as seen in Figure 4.

$$y = \sigma\left(N_f(x)\right) \times x \tag{3}$$

Here, the input feature map is represented by $x$, and the output feature map is represented by $y$. The SAM block's output is computed using a non-linear function denoted by $N_f$. The attention function, $\sigma\left(N_f(x)\right)$, is responsible for assigning a value between 1 and 0 to the spatial features of the input $x$ based on their priority level. Specifically, higher priority spatial features receive a value closer to 1, while lower priority spatial features receive a value closer to 0.

### 3.4. Bounding Box Regression

To simplify the intricate task of object detection, it can be broken down into two sub-tasks: object classification and object localization. Object localization employs Bounding Box Regression (BBR) to precisely locate the object of interest within an image and to generate predictions of a rectangular bounding box surrounding it. The predicted bounding box is then compared to the ground truth bounding box to calculate the Intersection over Union (IoU) [13] loss, which represents the overlap area between the two boxes. IoU, also known as the Jaccard Index, is a commonly used metric for measuring the similarity and diversity between two arbitrary shapes. It is calculated as the ratio of the intersection and union of the predicted bounding box ($A$) and the ground-truth bounding box ($B$), and can be expressed mathematically as follows:

$$IoU(A,\ B) = \frac{A \cap B}{A \cup B} \tag{4}$$

The use of IoU loss in BBR is common, but it fails when the predicted bounding box and ground truth bounding box fail to overlap (i.e., when $IoU(A,\ B) = 0$). To address this issue of disjoint $A$ and $B$ in IoU, the proposed MobDet3 object detector incorporates the Generalized IoU (GIoU) [13] loss. By iteratively adjusting the position of the predicted bounding box toward the ground truth box, the GIoU metric enhances the overlap area between the two boxes. This effectively resolves the issue of disjointedness encountered in the standard Intersection over Union (IoU) measurement. The GIoU loss is expressed mathematically as follows:

$$\mathcal{L}_{GIoU} = 1 - IoU + \frac{|C - A \cup B|}{|C|} \tag{5}$$

In this equation, $C$ represents the smallest box that encompasses both $A$ and $B$. The experiments conducted in [13] reveal that GIoU surpasses Mean Squared Error (MSE) and IoU losses in performance. These experiments also demonstrate the effectiveness of GIoU in addressing the problem of vanishing gradients that arise in cases where objects do not overlap.

## 4. NXP BlueBox 2.0 with RTMaps: A Development Platform for Automotive High-Performance Computing (AHPC)

This section presents a concise introduction to the NXP BlueBox 2.0 hardware that we utilize to implement the trained weights. The NXP BlueBox is a smart edge development platform developed for self-driving (autonomous) vehicles.

### 4.1. A Brief Overview of NXP BlueBox 2.0

The NXP BlueBox was developed by NXP Semiconductors in 2016 as an intelligent edge development and sensor-fusion platform for use in automotive applications, such as autonomous or self-driving vehicles. The first generation of NXP BlueBox, referred to as BlueBox generation 1, was initially launched at the NXP Technology Forum in Austin, Texas, USA [16]. Shortly after, NXP introduced BlueBox generation 2 (called BlueBox 2.0), which includes three new ARM-based processors for computer vision, high performance computing, and radar information processing: the S32V234, LS2084A, and S32R274, respectively. NXP BlueBox 2.0 is an Automotive High-Performance Compute (AHPC) development platform that adheres to rigorous automotive compliance and standards, such as ASIL-B/C and ASIL-D, to ensure its reliability and safety in all conditions and workloads.

#### 4.1.1. Computer Vision Processing with S32V234 (S32V)

The S32V234 processor is specialized in performing computer vision operations. It consists of a 4-core ARM Cortex A53 CPU that runs at 1.0 GHz, combined with another ARM Cortex M4 CPU for functional safety support. Additionally, it has a 4 MB internal SRAM and a 32-bit LPDDR3 controller to accommodate more memory when necessary. To comply with ASIL-B/C automotive standards, the S32V234 also has an onboard Image Signal Processor (ISP) that surpasses the requirements.

#### 4.1.2. High Performance Computing with LS2084A (LS2)

The LS2084A processor is tailored for high-performance computational tasks. It features an 8-core ARM Cortex A72 CPU operating at 1.8 GHz, and supports the LayerScape LX2 family, which is a group of multi-core processors designed for high-performance computing by NXP. Additionally, it is equipped with two 72-byte DDR4 RAMs. The LS2084A processor adheres to the Q100 Grade 3 automotive standards for reliability, providing 15 years of dependable performance.

#### 4.1.3. Radar Information Processing with S32R274 (S32R)

Designed for the real-time processing of radar information, the S32R274 processor is equipped with a dual-core Freescale PowerPC e200z4 CPU running at 0.12 GHz and an additional checker core. It also has B flash memory and 1.5 MB of SRAM. The processor is capable of performing radar information processing on-chip and is compliant with ASIL-D automotive standards for reliability and safety.

### 4.2. Real-Time Multi-Sensor Applications (RTMaps)

Intempora developed Real-time Multi-sensor applications (RTMaps) [42] as remote studio GUI-based software that enables the development of automotive applications for autonomous driving, motion planning, and Advanced Driver-Assistance Systems (ADAS) on NXP BlueBox 2.0. RTMaps can capture, view, and process data from multiple sensors in real-time and supports both Windows and Linux operating systems. Moreover, it is compatible with the PyTorch and TensorFlow machine learning frameworks. Figure 9 provides an overview of RTMaps and the NXP BlueBox 2.0 development system.

In the upcoming sections, we present our experiments that leverage open-source Python libraries and a real-time image classification algorithm created through the Python scripting language. We then deploy the algorithm onto NXP BlueBox 2.0 and draw conclusions based on the results.

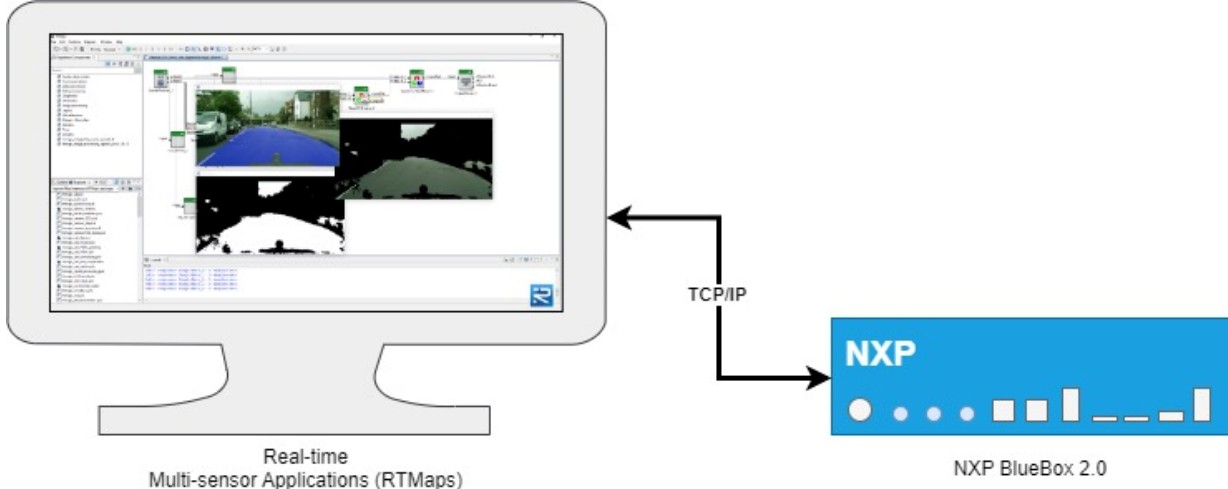

**Figure 9.** An overview of RTMaps and the NXP BlueBox 2.0 development system.

## 5. An Analysis of Experimental Results

In addition to identifying the location of an object within an image, a modern object detector possesses the ability to determine the specific type of object by enclosing it with rectangular bounding boxes. Typically, an object detector undergoes training using a dataset that consists of labeled images, commonly referred to as ground-truth values. These ground-truth values include pre-labeled images with class and object labels, accompanied by bounding boxes that precisely define the location of each object within the image.

### 5.1. Computer Vision Datasets

The proposed ModDet3 modern object detector's robustness is tested using two publicly available multi-class datasets: Berkeley DeepDrive BDD100K [14] and Microsoft COCO [15]. It encompasses annotated images and bounding boxes that are utilized for tasks, such as object detection and tracking the detected objects. These datasets cover a wide range of scenarios, including birds' eye view of the road for autonomous driving, everyday objects, and humans.

The datasets are divided into predetermined train and test sets, each consisting of a fixed number of images. Figure 10 provides a visual representation of sample images from the datasets employed to train and test ModDet3. A statistical comparison between the two datasets is provided in Table 1, showing their distinct features and characteristics, which enables a comprehensive evaluation of the proposed object detector.

**Table 1.** Dataset Statistics used to Training and Testing Purposes.

| Name of the Dataset | Data Type | Number of Classes | Number of Images | | Resolution of Images | Class Overlap |
|---|---|---|---|---|---|---|
| | | | Train | Test | | |
| BDD100K | Road | 10 | 70,000 | 20,000 | 1280 × 720 | True |
| COCO | Common | 80 [1] | 118,287 | 5000 | Multi-scale | True |

[1] The COCO dataset includes 80 classes of objects and 11 categories of "stuff" (background elements such as sky or grass) that may not have corresponding object labels.

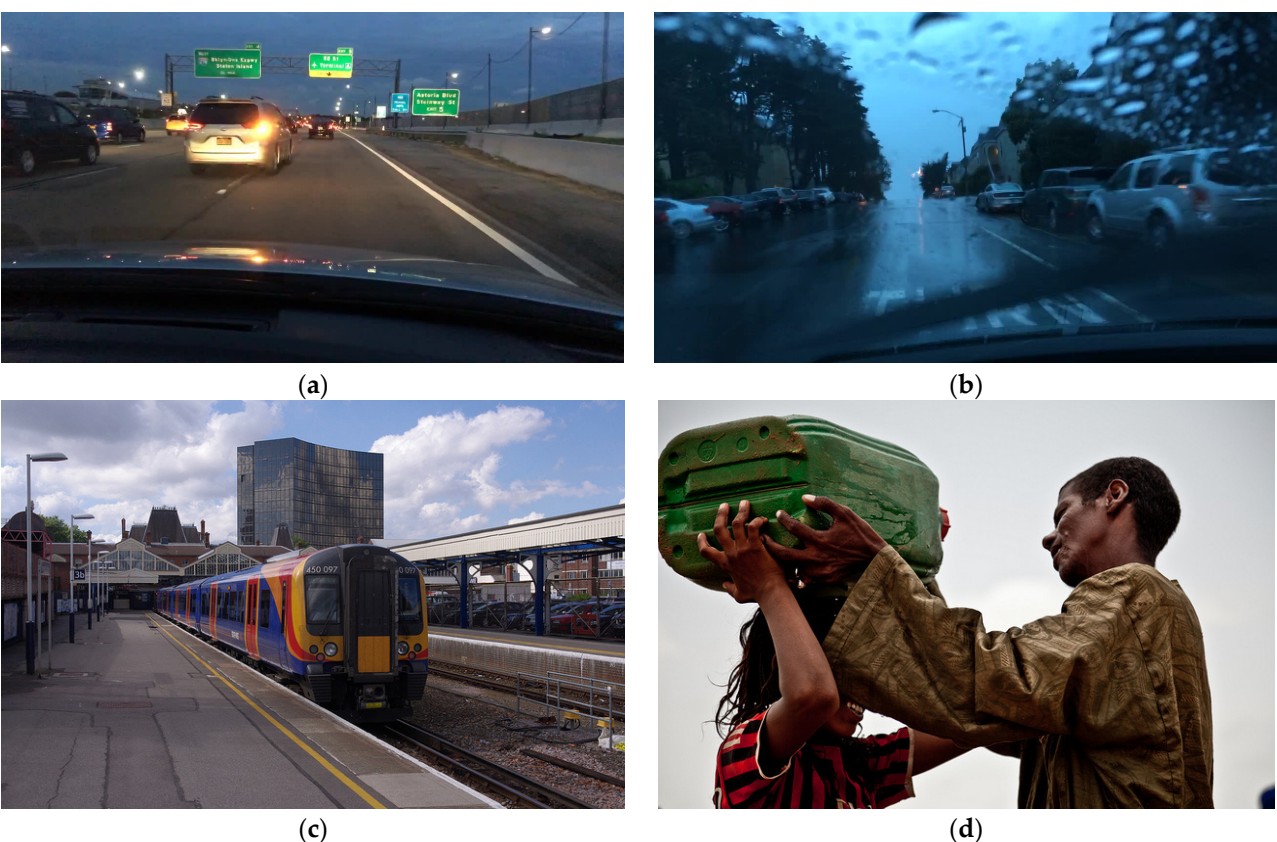

**Figure 10.** This figure offers a glimpse into the two datasets that were utilized for the experiments outlined in this paper. Images (**a**,**b**) showcase multi-class objects from the BDD100K dataset, while images (**c**,**d**) depict multi-class objects from the COCO dataset, respectively. The datasets consist of numerous images with diverse lighting conditions, complex scenarios, and objects belonging to different classes.

### 5.1.1. BDD100K Dataset

BDD100K (Berkley Deep Drive 100K) is a collection of 100,000 driving videos with a duration of 40 s each, gathered from over 50,000 rides across various regions, including the New York and San Francisco Bay Area, designed to assess the advancement of image recognition algorithms for autonomous driving.

This dataset is diverse, with a mixture of geographic, environmental, and weather conditions, which is beneficial in developing models that are capable of handling various scenarios. It contains a variety of street types, such as city streets, residential areas, and highways, and videos recorded during different times of the day and weather conditions. The dataset is divided into training (70,000 images), validation (10,000 images), and testing (20,000 images) sets, with the 10th second frame of each video annotated for image tasks and the entire sequences used for tracking tasks.

### 5.1.2. Microsoft COCO Dataset

The COCO (Common Objects in Context) dataset, introduced by Microsoft in 2014, is a widely used large-scale dataset for object detection, segmentation, and captioning tasks. The dataset comprises 328,000 images depicting everyday scenes, encompassing a total of 2.5 million labeled instances of common objects in their natural environment. This extensive dataset offers a diverse range of examples for training and evaluating object detectors.

This dataset includes 80 object classes, such as person, bicycle, car, and train, as well as 11 stuff object classes, such as grass, street, and sky, that provide contextual information. The inclusion of these diverse object and stuff classes offers challenging scenarios for evaluating the proposed object detector thoroughly.

### 5.2. Evaluation Metrics for the Model

A GIoU threshold of 0.2 is chosen using a grid search method and applied to all models and datasets discussed in this paper. An object detector is deemed successful in detecting the target objects when it fulfills the following two conditions; therefore, the proposed object detector, MobDet3, is evaluated using a two-step assessment procedure as described below:

1. The ratio of overlap between the predicted bounding box and the ground-truth bounding box is calculated using GIoU, as defined in (5). When the computed GIoU value surpasses a predetermined threshold, it indicates that the object detector has successfully detected the target object within the image.
2. Once it is confirmed that an object has been successfully detected in step 1, the matching of class labels between the predicted and ground-truth bounding boxes is carried out.

Various evaluation metrics are used to assess the performance of an object detector [43]. The Mean Average Precision (mAP) is a commonly utilized benchmarking metric, calculated as the average of the Average Precision (AP) scores for all classes. AP is determined as the area under the Precision-Recall (PR) curve, where Precision (P) represents the model's capability to correctly identify target objects (i.e., the percentage of accurate predictions), and Recall (R) reflects the model's ability to locate all positives among the corresponding ground truth values. The mathematical definitions of these metrics are as follows:

$$\text{Precision} = \frac{\text{TP}}{\text{TP} + \text{FP}} \tag{6}$$

$$\text{Recall} = \frac{\text{TP}}{\text{TP} + \text{FN}} \tag{7}$$

$$\text{Average Precision} = \frac{\text{sum of all P values}}{\text{number of objects}} \tag{8}$$

$$\text{Mean Average Precision} = \frac{\text{sum of all AP values}}{\text{number of classes}} \tag{9}$$

Within this context, a True Positive (TP) is recognized when the GIoU value between the predicted bounding box and the ground truth bounding box is equal to or exceeds a threshold. This threshold is commonly set between 50% and 95%. On the other hand, false positives (FP) denote incorrect detections where the GIoU value falls below the threshold. False negatives (FN) indicate that the ground truth was not detected.

### 5.3. Experimental Configuration

The object detection algorithm named MobDet3 is developed using the Python programming language and is based on the most recent version of the PyTorch [44] machine learning framework based on the Torch library at the time when this paper was submitted. In the training phase, a batch size of 16 is used, and the learning rate is reset after each epoch using cosine decay. The training process is conducted over a total of 500 epochs.

The experiments presented in this paper were trained on a single GPU using a node from the GPU partition of the Carbonate supercomputer. Carbonate is a high-performance computing cluster equipped with 24 GPU-accelerated Apollo 6500 nodes purposefully designed for deep learning research. The Research Technologies division at Indiana University facilitated and administered each node in Carbonate's GPU partition. This work was partially supported by Shared University Research grants from IBM Inc. to Indiana University and funding from Lilly Endowment Inc. through its support for the Indiana University Pervasive Technology Institute [45]. Each node of Carbonate's GPU partition consists of:

- 3B NVIDIA Tesla V100 GPU

- 20-core Intel 6248 CPU
- 1.92 TB Solid-State Drive (SSD)
- 768 GB of RAM
- PyTorch 2.0.0
- CUDA 11.8
- Python 3.9

*5.4. Experimental Results*

This paper includes ablation studies that involve removing or replacing specific components of the proposed deep neural network to gain insights into its performance. To evaluate the importance of each module, a total of six versions of the MobDet3 network were developed, trained, and assessed using the two datasets. Table 2 provides a comprehensive overview of the distinctive characteristics of each version of the object detector utilized in this experimental analysis.

**Table 2.** List of Acronyms for Object Detection Architectures Used in Training and Ablation Study.

| Architecture Name | Description |
|---|---|
| MobDet3 | The architecture presented in this paper, known as MobDet3. |
| MobDet3 + FPN | MobDet3 solely incorporates the FPN network. |
| MobDet3-SAM | MobDet3 without spatial attention modules. |
| ShuffDet + FPAN | MobDet3 with a ShuffleNet backbone and a PAN network. |
| ShuffDet + FPN | MobDet3 with a ShuffleNet backbone and a FPN network. |
| SqDet + FPAN | MobDet3 with a SqueezeNet backbone and a PAN network. |
| SqDet + FPN | MobDet3 with a SqueezeNet backbone and a FPN network. |

In addition to the ablation study, qualitative performance comparisons were conducted with other lightweight PyTorch networks, such as ShuffleNet and SqueezeNet, and integrated into the proposed MobDet3 network as backbones. These comparisons, outlined in Tables 3 and 4, aim to effectively understand how the proposed approach qualitatively stacks up against other state-of-the-art architectures in object detection tasks without focusing on specific quantitative comparisons with any single baseline architectures, such as YOLOv5.

**Table 3.** An Overview of Performance Comparison on the BDD100K Dataset.

| Architecture Name | Backbone | Mean Precision | mAP | AP$_{50}$ | AP$_{75}$ |
|---|---|---|---|---|---|
| MobDet3 | MobileNetv3 | 58.30% | 31.30% | 45.36% | 33.94% |
| MobDet3 + FPN | MobileNetv3 | 54.68% | 30.21% | 41.85% | 32.79% |
| MobDet3-SAM | MobileNetv3 | 56.23% | 30.76% | 42.46% | 33.05% |
| ShuffDet + FPAN | ShuffleNet | 54.92% | 27.02% | 39.11% | 29.28% |
| ShuffDet + FPN | ShuffleNet | 50.81% | 25.83% | 37.81% | 28.02% |
| SqDet + FPAN | SqueezeNet | 51.61% | 26.66% | 38.52% | 28.90% |
| SqDet + FPN | SqueezeNet | 49.89% | 25.47% | 37.15% | 27.59% |

The combinations of ShuffleNet and SqueezeNet with the proposed Feature Pyramid Networks (FPN) and Pyramid Attention Networks (PAN) were named ShuffDet + FPAN and SqDet + FPAN, respectively. Moreover, ShuffDet + FPN and SqDet + FPN were used to represent ShuffleNet and SqueezeNet, with only FPN in the neck design.

**Table 4.** An Overview of Performance Comparison on the COCO Dataset.

| Architecture Name | Backbone | Mean Precision | mAP | AP$_{50}$ | AP$_{75}$ |
|---|---|---|---|---|---|
| MobDet3 | MobileNetv3 | 56.08% | 51.68% | 72.67% | 56.29% |
| MobDet3 + FPN | MobileNetv3 | 54.45% | 50.58% | 70.91% | 54.65% |
| MobDet3-SAM | MobileNetv3 | 54.87% | 50.91% | 71.58% | 55.07% |
| ShuffDet + FPAN | ShuffleNet | 49.87% | 45.53% | 64.45% | 52.57% |
| ShuffDet + FPN | ShuffleNet | 48.28% | 45.25% | 63.22% | 47.47% |
| SqDet + FPAN | SqueezeNet | 45.75% | 45.09% | 63.16% | 47.20% |
| SqDet + FPN | SqueezeNet | 45.80% | 44.66% | 62.48% | 46.80% |

The results presented in Tables 3 and 4 demonstrate the outstanding performance of the proposed object detector, MobDet3, in monocular streaming perception on the BDD100K dataset and general-purpose multiclass object detection on the Microsoft COCO dataset. Additionally, it can be inferred that the elimination of certain techniques, such as attention modules, leads to a decline in MobDet3's performance. Similarly, replacing the MobileNetv3 backbone, which is ultra-efficient, with other popular lightweight CNNs such as ShuffleNet and SqueezeNet significantly reduces MobDet3's object detection performance. Hence, the experimental results suggest that the backbone and head components of an object detector are crucial in improving its detection performance, as they are responsible for locating objects in the input image and detecting/refining the location of the bounding box, respectively. This emphasizes the significance of these components over the neck component in the overall performance of the object detector.

By incorporating innovative and adapted techniques and strategies in the backbone, neck, and head, the proposed MobDet3 object detector achieves substantial enhancements in performance, as demonstrated in Tables 3 and 4. Moreover, Figure 11 showcases a heatmap illustrating the class activation mapping for an input image sourced from the COCO dataset. This heatmap highlights the significant regions crucial for object detection, obtained by analyzing the gradients of two objects as they pass through the final convolution layers of the proposed MobDet3 network.

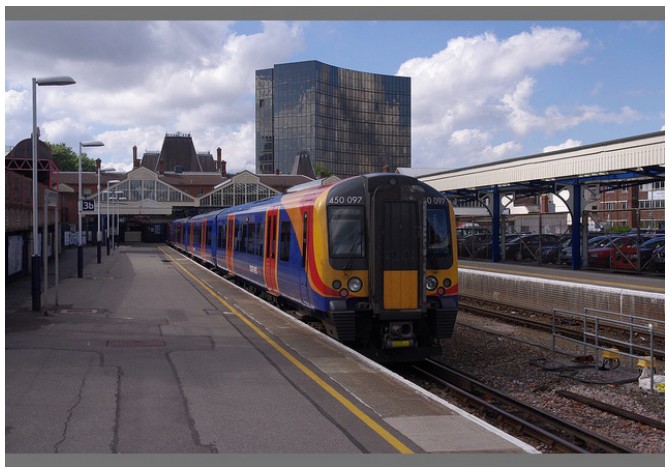
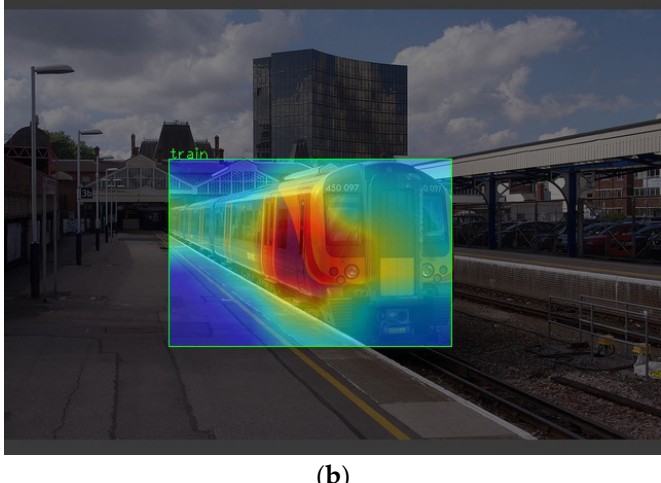

(**a**)   (**b**)

**Figure 11.** An illustration of the input image example shown in (**a**) and heatmaps of class activation mapping for the 'train' object shown in (**b**). The proposed MobDet3 object detection network utilizes the stronger activations indicated by the red regions of the heatmap for making predictions. The darker regions in (**b**) indicate points in the image where no activation occurred.

FPS, which stands for Frames Per Second, is an essential metric in object detection because it directly measures the speed or rate at which an object detection algorithm can process video frames. It refers to the number of individual images or frames that can be processed or analyzed by the algorithm in one second.

During this evaluation, a bird's eye view video stream, captured by a front camera, was fed into the RTMaps Studio software. This setup aimed to simulate a video stream obtained from a monocular front camera installed on a vehicle. The provided video underwent processing using the proposed MobDet3 network for the real-time detection of vehicular objects. The FPS results obtained from this analysis were then extrapolated accordingly. Figure 12 illustrates the RTMaps setup and evaluation utilizing NXP BlueBox 2.0, capturing a screenshot of the process.

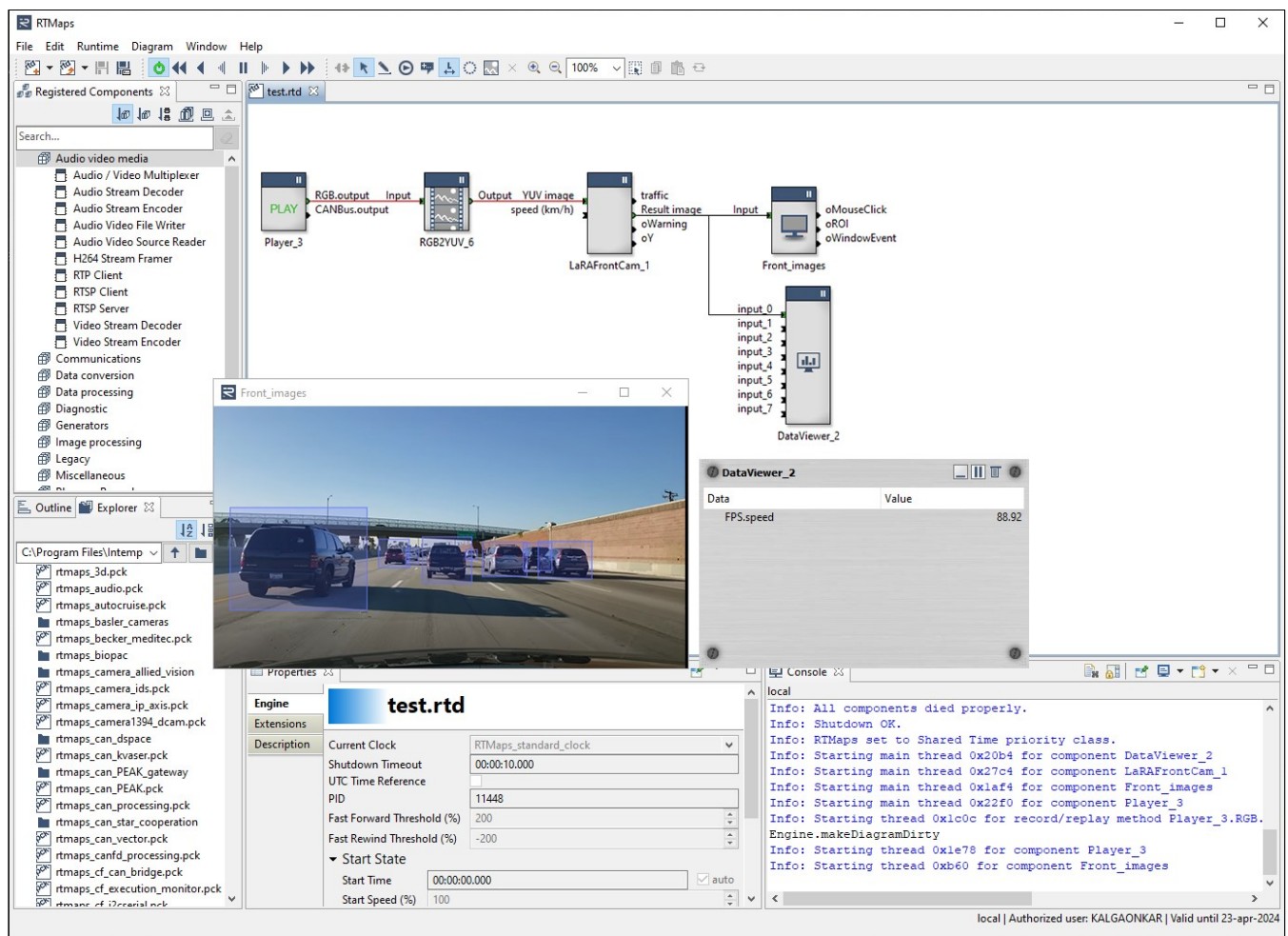

**Figure 12.** Evaluation of the proposed MobDet3 object detector's performance when deployed on NXP BlueBox 2.0 using RTMaps.

Table 5 presents a list of acronyms for various object detection architectures used in both the training and ablation studies. It includes the respective inference times of each architecture on the NXP BlueBox 2.0 platform, measured in Frames Per Second (FPS). Among the architectures evaluated, *MobDet3* demonstrates an impressive inference time of 88.92 FPS. Modifying MobDet3 with a Feature Pyramid Network (FPN) only slightly improves the inference performance, yielding 90.21 FPS at the cost of detection accuracy outlined in Tables 3 and 4. Comparatively, *ShuffDet + FPAN* and *ShuffDet + FPN* exhibit slightly lower inference times at 79.30 FPS and 80.92 FPS, respectively. Furthermore, *SqDet + FPAN* and *SqDet + FPN* demonstrate inference times of 75.29 FPS and 78.08 FPS, respectively.

**Table 5.** Inference Time on NXP BlueBox 2.0.

| Architecture Name | Inference Time on NXP BlueBox 2.0 |
|---|---|
| MobDet3 | 88.92 FPS |
| MobDet3 + FPN | 90.21 FPS |
| MobDet3-SAM | 87.47 FPS |
| ShuffDet + FPAN | 79.30 FPS |
| ShuffDet + FPN | 80.92 FPS |
| SqDet + FPAN | 75.29 FPS |
| SqDet + FPN | 78.08 FPS |

This section, thus, focuses on to present the results of the ablation study and qualitative comparisons with other state-of-the-art lightweight architectures, rather than a direct quantitative comparison against any specific baseline. These findings collectively demonstrate the effectiveness and significance of the proposed MobDet3 object detection architecture in various object detection tasks.

*5.5. Limitations of the Proposed Methodology*

Despite the promising results and superior inference time performance demonstrated by the proposed object detection architecture, MobDet3, there are potential limitations that should be acknowledged. First, the evaluation of the proposed network is conducted solely on the NXP BlueBox 2.0 platform, which may limit the generalizability of the findings to other hardware configurations or environments. Additionally, the dataset used for training and evaluation might have specific characteristics that could impact the performance of the model on different datasets with varying object distributions or scene complexities. Furthermore, the ablation study primarily focuses on removing certain modules, but it may not cover all possible architectural variations or combinations. As with any deep learning-based approach, the proposed methodology may also be susceptible to overfitting, requiring careful tuning of hyperparameters and regularization techniques. Despite these limitations, the work presented in this paper serves as a valuable contribution and lays the groundwork for future research in refining and extending object detection architectures for real-time applications.

**6. Conclusions**

In this paper, we introduce a contemporary object detection network designed to effectively detect objects belonging to multiple classes within an image. Our proposed network incorporates a customized variant of the efficient MobileNetv3 CNN as the backbone. Additionally, we integrate a path aggregation network with a modified feature pyramid network and spatial pyramid pooling in the neck module to further enhance the performance of the object detection process. Moreover, the architecture incorporates modified spatial attention module blocks for attentive feature aggregation, which constitutes a novel design aspect of the head module. To assess the performance of the proposed object detector, extensive evaluations are conducted on two widely used multi-class datasets: BDD100K and COCO.

The experiments encompass comprehensive analysis and benchmarking of various PyTorch versions of lightweight CNNs integrated as backbones within the architecture of MobDet3. Furthermore, MobDet3 is evaluated on the NXP BlueBox 2.0 edge development platform for autonomous vehicles to validate its real-time inference performance. The proposed object detector achieves notable performance, demonstrating good interpretability and robustness. Based on the experimental findings, it can be concluded that both the backbone and head components of an object detector significantly contribute to enhancing its detection performance. The proposed object detector successfully combines various techniques and strategies, proving its effectiveness in achieving real-time object detection on edge devices with limited computational resources.

**Author Contributions:** Conceptualization, P.K. and M.E.-S.; methodology, P.K.; software, P.K.; validation, P.K. and M.E.-S.; writing—original draft preparation, P.K.; writing—review and editing, P.K. and M.E.-S.; supervision, M.E.-S. All authors have read and agreed to the published version of the manuscript.

**Funding:** This research received no external funding.

**Data Availability Statement:** Openly available public datasets, such as BDD100K and COCO, have been utilized in the study. They are cited in references [14,15] of this paper.

**Acknowledgments:** The authors would like to acknowledge the Indiana University Pervasive Technology Institute for providing supercomputing and storage resources, as well as the Internet of Things (IoT) Collaboratory at the Purdue School of Engineering and Technology at Indiana University and Purdue University at Indianapolis, which have contributed to the research results reported within this paper.

**Conflicts of Interest:** The authors declare no conflict of interest.

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
