# Peer review of "An Improved Lightweight Network Using Attentive Feature Aggregation for Object Detection in Autonomous Driving"

_jlpea, doi:10.3390/jlpea13030049_

Round 1

Reviewer 1 Report

Well-written paper that does a thorough job of describing an improved object detection architecture using MobileNetv3 as a backbone.  The proposed method is appropriately validated against two large datasets, one of which is specific to self-driving vehicles.  The results show significant improvement over other approaches.  

One thing I found missing is a performance comparison against other state-of-the-art object detection architectures trained on the same datasets.  An ablation study is provided, which shows how performance degrades with respect to certain parameters of the architecture; however, it is not clear how well the entire method compares to a well-known baseline, for example, YOLOv5.  Is this somehow implicit in the ablation study?

I also find it odd to characterize the neural network architecture in terms of frames per second in the abstract, as this metric is very much hardware-specific.  This metric is listed in the abstract without any reference to the specific hardware used.  While it certainly makes sense to report the FPS on the hardware used, it does not make sense to tout an FPS metric as a property of the neural network architecture itself.

Reviewer 2 Report

The paper is well written and organized. The proposed methodology is properly justified highlighting its promising results via the experiments.                Major revisions:

- Please provide the training time for all the comparisons / experiments in section 4. 

- Please provide the inference time of the proposed network compared to the ones that is compared in section 4. Does the proposed network provides better inference time and performance/metrics?

- Please add a short paragraph in which you provide possible limitations of the proposed methodology.  

Minor comment:

- Mathematical definitions (6) and (7): is necessary the "TP/all detections"? 

Round 2

Reviewer 2 Report

Major concerns were addressed thus the paper is ready for publication